# Evaluation of the Feasibility of 2D-SWE to Measure Liver Stiffness in Healthy Dogs and Analysis of Possible Confounding Factors

**DOI:** 10.3390/ani13223446

**Published:** 2023-11-08

**Authors:** Ji’ang Pi, Eric Wenhao Foo, Xueyu Zang, Shuai Li, Yanbing Zhao, Yongwang Liu, Yifeng Deng

**Affiliations:** 1Department of Veterinary Clinical Science, College of Veterinary Medicine, Nanjing Agricultural University, Nanjing 210095, China; 2021807123@stu.njau.edu.cn (J.P.); 2021107206@stu.njau.edu.cn (E.W.F.); 2021807122@stu.njau.edu.cn (X.Z.); 2021207083@stu.njau.edu.cn (S.L.); 2Teaching Animal Hospital of Nanjing Agricultural University, Nanjing 210095, China; zybing@njau.edu.cn (Y.Z.); lyw@njau.edu.cn (Y.L.)

**Keywords:** canine, anesthesia, diagnostic imaging, liver

## Abstract

**Simple Summary:**

Currently, there is a lack of diagnostic methods for liver fibrosis in veterinary medicine. Two-dimensional shear wave elastography is a non-invasive diagnostic technique widely used in human medicine. However, there is a limited amount of relevant research in veterinary medicine, with only a few studies available. These studies suffered from small sample sizes, and the results of different experiments contradict each other. The aim of this study was to measure the range of liver stiffness in healthy dogs and to investigate the factors influencing it. We found that liver stiffness in healthy dogs was 3.96 ± 0.53 kPa, which was influenced by anesthesia and measurement site. These findings provided a theoretical basis and data support for its application in veterinary clinical practice. At the same time, the determination of health values contributed to the subsequent study of two-dimensional shear wave elastography in dogs.

**Abstract:**

(1) Background: Two-dimensional shear wave elastography (2D-SWE) is a non-invasive method widely used in human medicine to assess the extent of liver fibrosis but only rarely applied to veterinary medicine. This study aimed to measure liver stiffness in healthy dogs and investigate the factors that impacted 2D-SWE measurement. (2) Methods: In total, 55 healthy dogs were enrolled and subjected to 2D-SWE measurements before and after anesthesia administration. Post-anesthesia 2D-SWE measurements and computerized tomography (CT) scans were obtained. (3) Results: The liver stiffness range in healthy dogs was 3.96 ± 0.53 kPa. In a stratified analysis based on confounding factors, liver stiffness was influenced by measurement site and anesthesia, but not by sex. No correlation was observed between liver stiffness and weight or liver CT attenuation. (4) Conclusions: 2D-SWE is feasible for liver stiffness measurement in dogs. Anesthesia and measurement site are sources of variability. Therefore, these factors should be considered while recording 2D-SWE measurements. Our data on liver stiffness in healthy dogs can serve as the basis for future studies on 2D-SWE to assess pathological conditions in dogs.

## 1. Introduction

Liver stiffness is one of the most important indicators of liver disease, as changes in liver stiffness often accompany the progression of liver disease [1]. Studies have shown that approximately 12% of dogs have varying degrees of hepatic fibrosis [2]. Liver fibrosis is a key factor leading to changes in liver stiffness, and different levels of liver stiffness correspond to different degrees of liver fibrosis [3].

For this reason, a non-invasive imaging diagnostic technique, elastography, was first introduced in human medicine. Elastography includes strain elastography (SE), acoustic radiation force impulse imaging (ARFI), transient elastography (TE), point shear wave elastography (p-SWE), and 2D-SWE. The type of force application is one of the crucial factors in the categorization of elastography, and it can be further divided into two categories, quasistatic and dynamic excitation methods, based on the method of force application. In the quasistatic excitation method, pressure is applied to the target tissue using a transducer or physiological motion to induce deformation, as in SE. On the other hand, the dynamic excitation method generates stimulation either mechanically on the body surface or using acoustic radiation force internally, as in TE and 2D-SWE [4,5,6].

2D-SWE is a promising technology to measure liver stiffness. As one of the earliest elastography techniques, SE can only provide qualitative or semi-quantitative measurements of tissue stiffness. It assesses the elasticity of the target tissue by comparing it to the surrounding healthy tissue. The application of acoustic radiation force has made it possible to quantitatively measure tissue stiffness values. TE, as a shear wave imaging technique, can only measure the average hardness along a single measurement line. In contrast, 2D-SWE not only allows quantitative measurement of liver stiffness but also allows the placement of multiple regions of interest (ROI) in the target area, thereby expanding the measurement range and providing more representative data. 2D-SWE integrates conventional ultrasound and color-coded tissue stiffness in real time to effectively avoid the inclusion of non-targets, thereby producing more reliable results. 2D-SWE allows real-time placement of the ROI under the guidance of conventional ultrasound, avoids blind inspection, and provides improved accuracy while reducing operational difficulties. Studies have shown that an operator can be considered a 2D-SWE expert after 300 cases of conventional ultrasound experience or more than 50 cases of elastography experience. Moreover, the use of 2D-SWE is not limited by the presence of ascites. 2D-SWE is an excellent diagnostic imaging technique and offers promising clinical applications in veterinary medicine [7,8,9].

As a form of shear wave imaging, 2D-SWE directly measures the physical quantity of the shear wave velocity itself. The process of performing 2D-SWE measurements typically begins through the application of an acoustic radiation force to multiple points within a larger ROI. The arrival time of shear waves generated at each focal point along a transverse line is then determined. This collected information is used to calculate the shear wave velocity (SWV, C, m/s), which is then automatically converted to Young’s modulus (E, kPa) in most cases using the equation E = 3ρC^2^, where ρ is the tissue density, defined as 1.00 kg/m^3^ [10,11,12].

The measurement results are encoded by the ultrasound machine into color images that can be overlaid on B-mode grayscale images or displayed side-by-side with B-mode images. The transparency and color display scale can be adjusted. By placing the ROI at the desired location, quantitative statistical data of Young’s modulus of the target tissue, such as mean, standard deviation, minimum and maximum values, can be obtained [11,13].

Liver biopsy is still considered as the gold standard for diagnosis of liver diseases, but its invasive nature can lead to complications such as bleeding, pain, and other complications [14]. Imaging diagnostic tools such as ultrasound elastography and CT angiography have increasingly played an important role in the diagnosis of liver diseases [15,16]. As non-invasive imaging technologies, ultrasound elastography techniques such as two-dimensional shear wave elastography (2D-SWE) have widespread applications in human medicine. Published studies have confirmed its good sensitivity and specificity [17,18]. CT, as an advanced diagnostic imaging technology, offers advantages such as non-invasiveness and high sensitivity, which are critical properties in the diagnosis of various liver diseases [19]. However, this technique exposes the patient to the risks associated with radiation and anesthesia, especially in canine patients with liver disease [20]. In comparison with liver biopsy and CT, elastography serves as a non-invasive, harmless technique that is highly reproducible and can qualitatively as well as quantitatively diagnose liver fibrosis [21].

There is limited published literature on the use of 2D-SWE in veterinary medicine and only a few articles have investigated its feasibility and repeatability in measuring canine liver stiffness. Recently published 2D-SWE studies in veterinary medicine have failed to fully investigate confounding factors such as measurement site, state of anesthesia, and the effect of spontaneous breathing, and to measure liver stiffness in healthy dogs. Further, the results of these studies are contradictory, owing to small sample sizes [2,22,23,24].

Herein we aimed to measure liver stiffness in a sufficient number of healthy dogs to establish a range of healthy values and to aid in the diagnosis of liver fibrosis and other liver diseases in dogs. Identifying an appropriate range of liver stiffness values in healthy dogs will facilitate future research on the use of the 2D-SWE technique in dogs. To this end, we determined the feasibility of 2D-SWE to measure liver stiffness in 55 healthy dogs and investigated the effects of several factors on 2D-SWE measurements. Previous studies have shown that 2D-SWE had good applicability and accuracy in diagnosing liver disease in human medicine [3,25]. In addition, some studies showed that 2D-SWE was also feasible in dogs [2,22]. Therefore, this study aimed to use 2D-SWE to measure liver stiffness in healthy dogs and hypothesized that factors such as measurement site, anesthesia, weight, and sex might influence liver stiffness. We hypothesized that there was a correlation between liver stiffness and liver CT attenuation.

## 2. Materials and Methods

### 2.1. Animals

Fifty-five healthy dogs (26 male and 29 female) were included in this study. There were 19 mixed-breed dogs, 28 spaniels, 1 German shepherd, 1 labrador, 3 reindeers, and 2 beagles, with an average age of 3 years (range 1–4 years) and a median weight of 13 kg (range 4.5–32.5 kg). All dogs were deemed to be healthy, as per physical examination, routine blood tests, blood biochemistry (Appendix A), abdominal CT scan, and echocardiogram. Dogs with abnormal test results were excluded from the study. Experimental dogs with ascites on B-mode ultrasound or liver abnormalities were excluded from the study. Prior to their inclusion in the study, all dogs were required to undergo a body condition score (BCS), and only those with a score of 3 (3/5) were enrolled in the study. Dogs were housed at a constant temperature of 26 °C with one dog per cage space, and fed commercial food (Singen BD80, Shanghai Singen Animal Health, Shanghai, China) and clean water. All experiments were conducted during the same anesthesia event. To control for data quality, all SWV measurements were performed by a single operator with over 2 years of ultrasound diagnostic experience. However, the operator was unaware of the ultrasound or blood test results.

This study was approved by the Ethics Committee of Nanjing Agricultural University.

### 2.2. Methods

Liver stiffness in dogs was measured using 2D-SWE in the awake state. Intravenous anesthesia was administered, and 2D-SWE measurements and abdominal CT scans were performed. Liver stiffness and CT attenuation were measured six times on each side of the liver lobes when 2D-SWE measurements and CT scans were completed. All CT scans and 2D-SWE measurements were performed in the same anesthesia event.

#### 2.2.1. SWE Measurement

2D-SWE measurements were obtained for each experimental dog in both the awake and anesthetized states.

Prior to examination in the awake state, dogs were fasted for more than 12 h, and 2D-SWE measurements were performed using a Philips ultrasound scanner (Philips Epiq7, Philips, Washington, DC, USA) and a c5-1 probe (1–5 mHz), as previously described [15,26,27,28,29]. Abdominal hair was clipped, and it was ensured that the dogs rested for at least 20 min before the ultrasound examination to avoid over-excitement. A sufficient amount of coupling agent was applied to the skin to facilitate an adequate fit between the probe and the skin. The probe was placed parallel to the rib cage, with its tip being perpendicular to the surface of the target organ. Minimal pressure was applied on the skin, which prevented changes in tissue stiffness caused by external forces. During measurements, the dog’s muzzle was pinched at the end of exhalation to force it to hold its breath for 2–3 s, which could reduce the effect of respiratory movements.

To maintain uniformity in measurement results, the ultrasound parameters for 2D-SWE were set to the system’s default configuration and no adjustments were made during the measurement process. The dual-screen mode of 2D-SWE was selected on the ultrasound machine, which displayed the confidence map on the left and the stiffness map on the right. The confidence map converted the confidence level of the measurements into different colors, which were overlaid on the ultrasound greyscale map. Green was regarded as highly reliable data, while red indicated unreliable data. The stiffness map represented tissue stiffness, with blue and red indicative of the lowest and highest stiffness, respectively (Figure 1). The absence of a color overlay on the stiffness map suggested that the measured value had a low confidence level that could not be measured. A 2 × 2 cm fan-shaped elastic sampling frame was placed over the target tissue in B-mode images. The intrusion depth was 5 cm, and the sampling frame was set at a minimum of 10 mm below the liver capsule to avoid possible reverberation artifacts. Then, a circular ROI of 10 mm diameter was placed in the flexible sampling frame. Care was taken when placing the elastic sampling frame and ROI to avoid blood vessels, bile ducts, and other membranous structures. The ROI placement also needed to meet additional two requirements. First, the color needed to remain stable on the stiffness map for a few seconds. Second, it needed to be green on the confidence map. After placing the ROI, the average, median, maximum, and interquartile range to the median ratio (ICR/MED) of stiffness within the ROI area were measured. According to the liver elastography guidelines originally developed from TE in human medicine, only an ICR/MED less than or equal to 30% was considered statistically significant, and the average represented the stiffness of the tissue within the ROI [5,30]. Tissue stiffness was measured as SWV, which was automatically converted to Young’s modulus by an embedded software package. 

The 2D-SWE measurements were conducted in the left lateral and right lateral positions to measure the liver stiffness of the left and right lobe, respectively. Measurements were repeated with different frames and ROIs until six valid observations were obtained. The mean values of SWEs were representative of the liver modulus values.

#### 2.2.2. Anesthesia

All animals were anesthetized with an intravenous dose of xylazine hydrochloride (0.85 mg/kg), tiletamine (4 mg/kg), and midazolam (0.2 mg/kg). During the anesthesia, heart rate, blood pressure, oxygen saturation, and end-tidal CO_2_ were continuously monitored.

#### 2.2.3. CT Scan

A Hitachi CT scanner was used with the following scan parameters: slice thickness of 1.25 mm to 2.50 mm, tube voltage of 120 kV, tube current of 200 mA, rotation time of 0.75 s, pitch of 1.0625, and collimation of 1.25 × 16.

Animal positioning and CT scan results are shown in Figure 2. The dogs were placed on the examination bed in an anesthetized state. Both forelimbs were pulled forward and the chest was placed against the examination bed with the median sagittal plane perpendicular to the examination bed. The examination site was fed into the scanning frame. Six sampling frames were consecutively placed in the left and right liver lobes once the CT scan was completed. The sampling frame avoided the gallbladder and large blood vessels.

#### 2.2.4. Data Analysis

Data were presented as mean ± standard deviation (SD) and analyzed using independent samples *t*-tests.

The sample size was determined by pre-experimentation, experimental design, and conventional numerical settings using PASS 15 (a type 1 error (*p* = 0.05), 90% power). Four healthy dogs were used in the pre-experimentation stage. The mean difference in liver stiffness values for the experimental dogs before and after anesthesia was 2.71 kPa (4.05 kPa vs. 6.76 kPa), with standard deviations of 104% and 141% for liver stiffness before and after anesthesia, respectively. Based on this, PASS 15 calculated that a sample size of 15 animals would result in >90% power using an independent sample *t*-test with a 5% type 1 error. 

Statistical analyses were performed using a commercially available statistical software package (SPSS, version 25, IBM Corp., Armonk, NY, USA; and Excel (Version 2310 Build 16.0.16924.20054) Microsoft Corp., Redmond, WA, USA). All measurements were expressed as Young’s modulus, and the data were analyzed by the Shapiro–Wilk test. Correlation analysis was used to determine the relationship between weight and liver stiffness. A value of *p* < 0.05 was considered statistically significant.

## 3. Results

### 3.1. Influence of the Measurement Site on the Measurement of Liver Stiffness

The liver stiffness in healthy dogs was 3.96 ± 0.53 kPa. We compared the liver stiffness between the left and right lobes in the awake and anesthesia states. (Table 1 and Figure 3). No significant difference was observed in the stiffness values between the left and right lobes in the awake state (*p* > 0.05). However, the liver stiffness was significantly higher in the left lobe than in the right lobe in the anesthesia state (*p* < 0.01).

### 3.2. Influence of Anesthesia on the Measurement of Liver Stiffness

Anesthesia significantly altered the stiffness of the liver in healthy dogs (Table 2 and Figure 4). The stiffness was significantly higher in the anesthesia state than in the awake state, both on the right and on the left liver lobes (*p* < 0.01 for the right liver lobe, *p* < 0.01 for the left liver lobe). The color changes in the stiffness map reflect the impact of anesthesia on liver stiffness in healthy dogs. After anesthesia, the color of the stiffness map shifted from a uniform light blue to a mixture of blue and green (Figure 5).

### 3.3. Influence of Sex on the Measurement of Liver Stiffness

Sex did not significantly affect liver stiffness in healthy dogs in either the awake or anesthesia states (Table 3). No significant difference was observed in the liver stiffness between male and female dogs in both awake and anesthesia states (*p* > 0.05 for the awake state, *p* > 0.05 for the anesthesia state).

### 3.4. Correlation Analysis of Weight and Liver Stiffness

We conducted a correlation analysis to explore the effect of weight on liver stiffness (Figure 6). The results revealed no significant correlation between weight and liver stiffness, either in the awake or anesthesia state (*p* > 0.05 for the awake state, *p* > 0.05 for the anesthesia state).

### 3.5. Correlation Analysis of CT Attenuation and Liver Stiffness

We determined the correlation of liver CT attenuation with liver stiffness (Table 4 and Figure 7). No significant correlation was found between liver CT attenuation and liver stiffness (*p* = 0.637 for the left liver lobe, *p* > 0.05 for the right liver lobe, and *p* >0.05 for the entire liver).

## 4. Discussion

In the present study, we investigated the applicability of 2D-SWE to assess liver stiffness in healthy dogs. The effects of anesthesia, measurement site, sex, and weight on the results were investigated, and the correlation between liver CT attenuation and liver stiffness was explored. The aim was to measure liver stiffness in a sufficient number of healthy dogs to establish a range of healthy values and to aid in the diagnosis of liver fibrosis and other liver diseases in dogs.

Unlike humans, dogs were not cooperative during examinations. Anesthesia is occasionally necessary in veterinary medicine to perform liver 2D-SWE. Tamura et al. identified motion as an important limitation in 2D-SWE measurements in dogs [31]. 2D-SWE should be performed in combination with sedation to reduce motion during data acquisition. Studies in veterinary medicine have reported inconsistent results on the effects of anesthesia on liver stiffness [2,22,23], probably owing to the differences in the anesthesia protocols used.

In our study, the liver stiffness of anesthetized dogs was significantly higher than that of awake dogs. This result is similar to a previously published study but different from the results of another study [2,23]. This discrepancy may be related to the choice of anesthesia protocol. One of these studies used propofol to induce anesthesia and isoflurane to maintain anesthesia. Propofol may increase liver perfusion in dogs, whereas isoflurane can have the opposite effect [32,33]. The counteractive effects of propofol and isoflurane might maintain liver stiffness to a steady value. Dogs were sedated with a combination of dexmedetomidine, methadone, and ketamine in another study. There is no published study that evaluated the effects of methadone and ketamine on liver perfusion. However, one study showed that dexmedetomidine caused perfusion modification in the liver [34]. The liver has a poorly distensible capsule, which may contribute to the increase in liver stiffness in response to an increase in perfusion. Dogs in this study were sedated with intramuscular zolazepam hydrochloride-tiletamine hydrochloride and medetomidine hydrochloride. The increase in the liver stiffness in the anesthesia state observed in our study might be related to the increase in liver perfusion. Previous studies have shown that xylazine hydrochloride did not significantly increase liver perfusion, while midazolam slightly reduced liver perfusion [35,36]. Unfortunately, there are no studies investigating the effects of tiletamine on liver perfusion. Evidence suggests that ketamine can prevent liver ischemia, hence, tiletamine, a drug similar to ketamine, may also play a potential role in increasing liver perfusion [35]. The association of these drugs may cause an increase in liver stiffness. The results of this study confirmed no significant difference in stiffness of the right and left lobes in the awake state, which contradicts the previously published results [31]. However, it was difficult to directly compare the results, owing to the use of instruments from different companies. 

In our study, the stiffness of the right liver lobe was slightly higher than that of the left lobe, but there was no significant difference. Interestingly, a significant difference was observed in stiffness between the right and left liver lobes in the anesthesia state. Consistent with the above speculation, we believe that this observation was related to changes in the liver blood perfusion. The anesthesia protocol used in this study might lead to differential changes in perfusion between the left and right liver lobes. The comparison of the changes in the liver stiffness before and after anesthesia made it clear that the right lobe experienced a greater increase in stiffness than the left lobe. Given the relatively poor distensibility of the liver capsule, we speculate that this phenomenon is attributed to the greater increase in the blood perfusion to the right lobe. Unfortunately, there are no reported studies on this topic. Given the time constraints and other factors, we were unable to thoroughly investigate this issue, and it warrants future study.

The results of this study indicated that sex had no significant effect on liver stiffness in healthy dogs, and that body weight was not significantly correlated with liver stiffness. Thus, our results are applicable across different sexes and body weights of dogs. In clinical practice, there is no need for concern about the influence of these two factors on measurement results. However, it is important to note that the dogs included in this experiment had a BCS of 3, which eliminated the effect of obesity on the results. The applicability of these results to obese dogs, which are commonly encountered in clinical settings, needs to be confirmed. Unfortunately, there are no published reports investigating the effect of obesity on 2D-SWE measurements in canine livers, and further studies are warranted to investigate this.

The lack of correlation between liver CT attenuation and liver stiffness indicated that 2D-SWE technology cannot be replaced by CT. In addition, 2D-SWE has several advantages over existing clinical imaging diagnostics in veterinary medicine. First, it offers high sensitivity and specificity and has been considered one of the preferred non-invasive imaging techniques for the clinical assessment of liver fibrosis in human medicine [11,37,38]. Second, 2D-SWE is non-invasive and it can be repeated several times in a short period, which significantly improves the diagnostic accuracy. Third, it can be used to diagnose various soft tissue diseases. A small number of studies have confirmed that 2D-SWE can measure tissue stiffness in organs such as the spleen, kidneys, and prostate [22,39,40]. All of the above suggests that 2D-SWE is an excellent diagnostic imaging tool with great clinical potential in veterinary medicine.

The present study has some limitations. First, the proportion of large dogs in this study was low, which may have influenced our findings. Therefore, the results of this study may not apply to large dogs. Second, all the data were measured by a single imaging physician which did not allow for the assessment of inter-observer variability. Third, the dogs in this study were not histologically examined to determine the health status of the liver, and this was presumed by physical examination, blood test, CT scan, and ultrasound. However, the dogs were relatively young, and no clinical signs or abnormalities of illness were found.

## 5. Conclusions

Here we demonstrated the applicability of 2D-SWE to measure dogs’ liver stiffness with high reproducibility. Liver stiffness was measured in 55 healthy dogs, and the results provided statistical support for the use of 2D-SWE in veterinary medicine. The range of liver stiffness in healthy dogs established herein will be useful in the diagnosis of liver fibrosis and other liver diseases in dogs. In addition, our study lays the foundation for future studies using 2D-SWE to diagnose pathological conditions that increase liver stiffness in dogs.

Anesthesia was deemed as a source of variability in liver stiffness. However, the effect of anesthesia on the liver stiffness of healthy dogs might be related to the anesthesia protocol used. The measurement site also affected 2D-SWE values and this effect was also, at least in part, related to the state of anesthesia. Therefore, these two factors should be interpreted and considered while recording 2D-SWE measurements in dogs.

## Figures and Tables

**Figure 1 animals-13-03446-f001:**
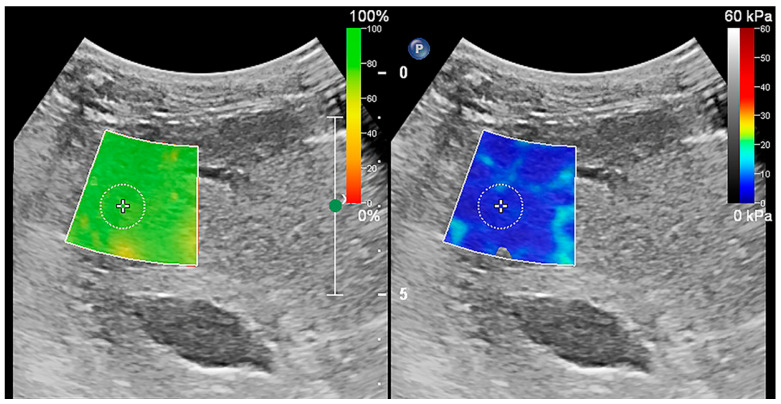
Shear wave elastography of the liver. Confidence map mode was on the left and stiffness map mode was on the right. In the confidence map mode, confidence levels were converted from low to high as red, yellow, and green. In the stiffness map mode, the liver stiffness was converted from low to high as blue, yellow, and red. The elastic sampling frame and ROI were placed to ideally avoid the vessels, bile ducts, and liver envelope, and the ROI was placed in the green position on the confidence map.

**Figure 2 animals-13-03446-f002:**
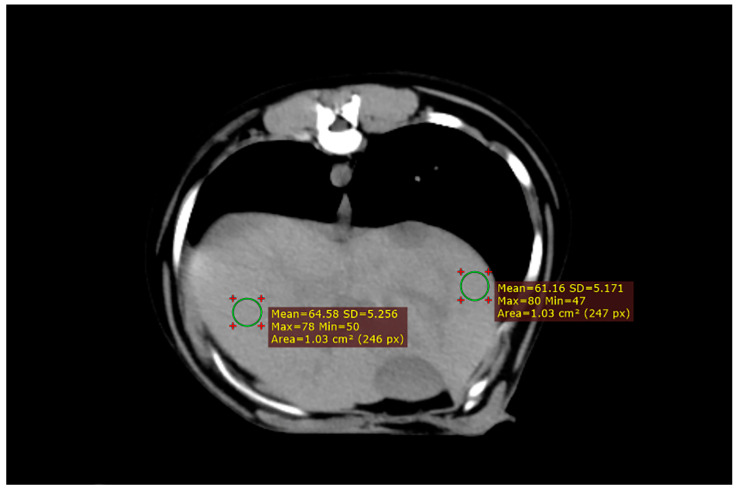
CT scan results. After completion of the CT scan, six sampling frames were placed on each side of the liver. It was important to avoid major blood vessels and bile ducts when placing the sampling frame to prevent measurement errors.

**Figure 3 animals-13-03446-f003:**
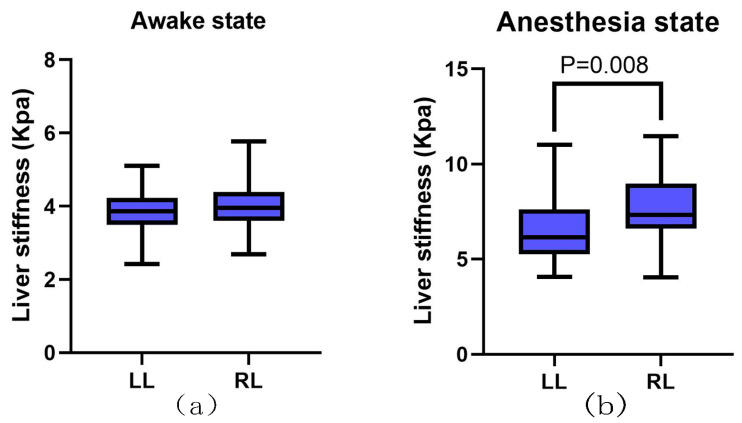
Effect of measurement site on liver stiffness values. (**a**) Stiffness of the left and right liver lobes in the awake state, (**b**) stiffness of the left and right liver lobes in the anesthesia state. The right lobe was significantly stiffer than the left lobe. The box extended from 25% to 75% percentile with the median and the whiskers extended to the limits of the data.

**Figure 4 animals-13-03446-f004:**
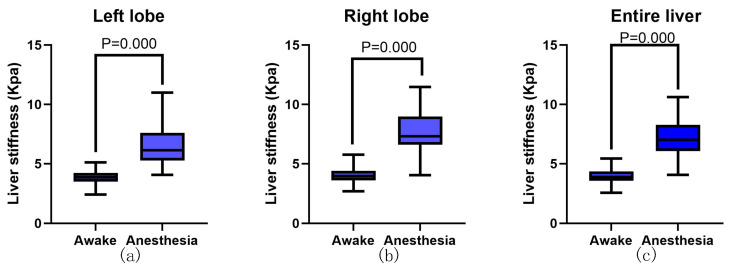
Effect of anesthesia on liver stiffness. (**a**) Stiffness of the left liver lobe in awake and anesthesia states. The left lobe was significantly stiffer in the anesthesia state than in the awake state (**b**) Stiffness of the right liver lobe in awake and anesthesia states, the right lobe was significantly stiffer in the anesthesia state than in the awake state. (**c**) After anesthesia, there was a significant increase in the liver stiffness of healthy dogs. The box extended from 25% to 75% percentile with the median and the whiskers extended to the limits of the data.

**Figure 5 animals-13-03446-f005:**
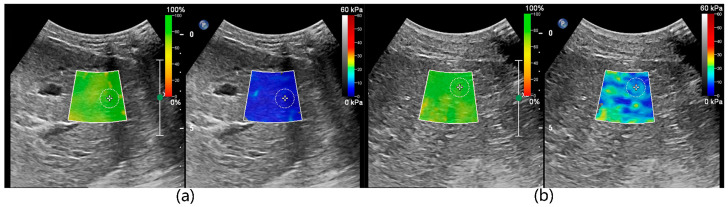
2D-SWE measurement in awake and anesthesia states state. (**a**) 2D-SWE measurements in the awake state (**b**) 2D-SWE measurements in the anesthesia states state. The color changes in the stiffness map reflect the effect of anesthesia on the liver stiffness. Before anesthesia, the stiffness map displayed a uniform blue color, while it exhibited a mixture of blue and green colors in the anesthesia state.

**Figure 6 animals-13-03446-f006:**
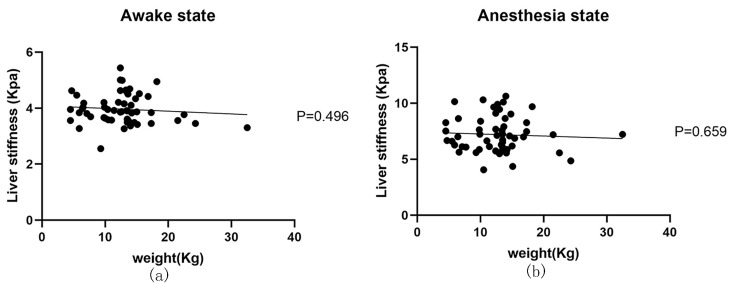
Correlation between weight and liver stiffness in different states. (**a**) Correlation analysis between weight and liver stiffness in the awake state revealed no significant relationship. (**b**) Correlation analysis between weight and liver stiffness in the anesthesia state also showed no significant relationship.

**Figure 7 animals-13-03446-f007:**
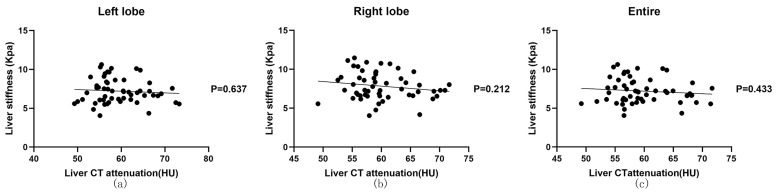
Correlation between liver CT attenuation and liver stiffness in the anesthesia state. (**a**) Correlation between liver CT attenuation and the stiffness of the left lobe. (**b**) Correlation between liver CT attenuation and stiffness of the right lobe (**c**) Correlation between liver CT attenuation and stiffness of the entire liver. No significant correlation was found between liver CT attenuation and liver stiffness.

**Table 1 animals-13-03446-t001:** Effect of measurement site on liver stiffness (n = 55).

	Awake (kPa)	Anesthesia (kPa)
Left lobe	3.88 ± 0.53	6.58 ± 1.11
Right lobe	4.04 ± 0.60	7.83 ± 1.75 **

** Indicates a highly significant difference from the left lobe. *p* < 0.05 indicates a significant difference, and *p* < 0.01 indicates a highly significant difference.

**Table 2 animals-13-03446-t002:** Effect of anesthesia on liver stiffness (n = 55).

	Awake (kPa)	Anesthesia (kPa)
Left lobe	3.88 ± 0.53	6.58 ± 1.11 **
Right lobe	4.04 ± 0.60	7.83 ± 1.75 **
Entire liver	3.96 ± 0.53	7.21 ± 1.57 **

** Indicates a highly significant difference from the awake state. *p* < 0.01 indicates a highly significant difference.

**Table 3 animals-13-03446-t003:** Effect of sex on liver stiffness in awake and anesthesia states (n = 55).

	Males (kPa)	Females (kPa)
Awake	3.99 ± 0.53	3.93 ± 0.53
Anesthesia	7.21 ± 1.82	7.20 ± 0.80

**Table 4 animals-13-03446-t004:** Results of the correlation analysis between liver CT attenuation and liver stiffness.

	CT Attenuation (HU)	LS (kPa)	R-Value
Left lobe	59.43 ± 5.62	6.58 ± 1.75	0.006
Right lobe	60.05 ± 5.06	7.83 ± 1.75	0.029
Entire liver	59.74 ± 5.12	7.20 ± 1.57	0.012

All data in the table were measured in the anesthesia state. LS = liver stiffness.

## Data Availability

We hereby declare that we have not created any new data in this study. Some experimental data are supported by the master’s and doctoral thesis database of Nanjing Agricultural University. However, some original data for ongoing study in the laboratory must be kept confidential. In addition, there are privacy concerns associated with some of the research data. For these two reasons, we are unable to fully disclose all data.

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
