# Peer review of "Evaluation of the Feasibility of 2D-SWE to Measure Liver Stiffness in Healthy Dogs and Analysis of Possible Confounding Factors"

_animals, 2023, doi:10.3390/ani13223446_

Round 1

Reviewer 1 Report

Comments and Suggestions for Authors

In the article "Confounding Factor Analysis of Liver Stiffness by 2D-SWE in 2 Healthy Dogs" the Authors present a novel and fascinating method of diagnosis of liver disease in dogs. The text lacks formating and order, which have to be changed for it to be published. The figures presented in the study are sufficient, but at least one figure of CT examination should be added.

Please format the text (and references) according to Animals MDPI submission guidelines 

line 24 - conducted

Discussion

Did you research the topic of ketamine and methadone influence on liver perfusion in other species? maybe it's worth comparing

Material and methods

Please add blood biochemistry results as supplementary material - since it's a liver study, this is important

What commercial feed were the dogs fed? When was the last meal before the examination?

The anesthesia protocol mentioned in the discussion section differs from one of the methods - did you use xylazine or zolazepam? please clarify

Please add the information on what CT scanner you used and what the parameters of the scan were

Also, at least one figure should contain pictures of CT scans - otherwise, please delete this method from the text

Author Response

Dear Reviewer,

We have carefully reviewed your comments and made the appropriate changes. Please see the attached document for detailed information on the changes.

Kind regards.

Reviewer 2 Report

Comments and Suggestions for Authors

Dear Authors,

thank you for submitting this interesting paper titled “Confounding Factor Analysis of Liver Stiffness by 2D-SWE in Healthy Dogs”.

The topic is potentially interesting for the readers, although some studies are already present in veterinary literature with a similar scope. Nevertheless, it could be potentially interesting, but I have major concerns that I would like to be addressed before recommending this paper for publication.

In general, I find the paper difficult to read, both for the overall way in which the paper is written and for the ideas that are not presented in a consistent way throughout the paper. For example, some sentences are written in present and others in past, I would suggest more uniformity to facilitate reading. Many sentences have been split in two with the use of punctuation, but this results in many sentences beginning with ‘and’ or ‘but’, please consider re-writing those. Some paragraphs of the discussion belong to the introduction and vice versa. I would consider other words instead of ‘the experiment’.

Furthermore, I would suggest adding more details regarding the imaging protocol, as key information are lacking in the materials and methods.

Introduction: Please provide references for all statements of the introduction. If no reference is available (because refers to personal belief or experience) than I would suggest avoiding putting them in this section and discuss them in the discussion.

All this section is also a bit confusing, as it does not introduce properly the aims of the study; the first paragraphs are also slightly redundant, the concept of CT and biopsies being suboptimal in diagnosis the fibrosis is repeated twice, please re-write.

The hepatic fibrosis is the focus of your introduction, but it is not the aim of your study; I would suggest re-writing the introduction on the basis of your study, healthy dogs. Then you can maybe briefly highlight the fact that normal ranges could help differentiate fibrosis or other diseases (check if more reports are present in the literature describing elastosonography of liver in dogs).

Please move the section of materials and methods before the results section.

Materials and methods: how did you chose to include 55 dogs? Was a power analysis performed?

I would suggest explaining better your technique both for US and for CT: the CT acquisition parameters are completely missing, the US only briefly explained. Was the exam standardized? Gain, focus, depth were all adjusted case by case or kept the same for uniformity? If the liver looked abnormal on US or CT, did you exclude the case?

I understand you measured the attenuation pre-contrast? How is attenuation in CT supposed to be correlated with the stiffness?

Results:

I would not divide ‘experimental results’ and ‘figures and tables’; figures and table should help explain the results and a new subsection is not needed.

From figure 3: I expect this to be the same dog in the same area pre and post-anesthesia?

But the location does not look the same, do you think having not standardized your region of interest could have altered your results? Not only your US image is different, you also put the ROI in two different locations at different depth.

Discussion:

I think the main points are not discussed here. You should find explanations on why some parameters differ and some others don’t. You discussed the anesthesia protocol and all other parameters are only briefly repeated from the results.

The different elasto techniques should be briefly summarized in the introduction, not in the discussion as this does not pertain to your results.

I would re-phrase the sentences where you used ‘Cha’s study or Puccinelli’s’ as normally a study involves more persons not only the first Author. It is also not really easy to read, I believe if you refer to it as ‘a study’ and ‘another study’ with the appropriate citation, it would be enough.

Lines 202-205: difference between right and left lobes of the liver may be related to the blood flow change of the liver. Can you explain the mechanism? In sedation/anesthesia some lobes are more perfused than others?

Did you consider the recumbency? Where the dogs scanned in lateral? Maybe first on the right side and then the left and that could influence the stiffness of the recumbent lobes?

Lines 212-224: This paragraph is redundant and also incorrect, as limitations of CT compared to US are already known before your study. Your results only prove that CT attenuation and stiffness as measured with elasto do not correlate. Your results do not say that US is better than CT, even more considering that your population refers only to healthy dogs.

Limitations: if the number is low, then you cannot say that anyway some things have been proved. Either the number is enough to trust the results or not.

Author Response

(The authors gave the same response as above.)

Round 2

Reviewer 1 Report

Comments and Suggestions for Authors

The authors answered all the question in a proper manner. For me, the article can be published in a current form.

Author Response

Dear Reviewer,

We have received your feedback on our submission. Thank you for your high evaluation of our manuscript and we wish you success in your future endeavors. We again extend our sincerest wishes to you.

Sincerely,

Deng Yifeng

dengyif@njau.edu.cn

Department of Veterinary Clinical Sciences, College of Veterinary Medicine, Nanjing Agricultural University, Nanjing, China

Reviewer 2 Report

Comments and Suggestions for Authors

Dear Authors, thank you for re-submitting the paper titled “Analysis of Confounding Factors of Liver Stiffness Measurement by 2D-SWE in Healthy Dogs”.

I appreciate the effort of answering my previous comments, but some major doubts remain. In particular, the introduction section did not change much in terms of contents, it is still largely based on canine hepatic fibrosis which is not the focus of your study.

I asked in my previous revision why you tried to correlate the attenuation in CT with the stiffness, as in my opinion these are two completely different factors. You replied that based on the formula it is obvious that they are related. Why are they not correlated then? I still think the pre-contrast attenuation is more similar to an echogenicity, if we want to parallel it with ultrasound, that has nothing to do with stiffness (otherwise we would not need elastography).

The Figures and Tables are not all necessary (see later in the comments); I also don’t think it is necessary to include ‘Note’ in the figure legends.

The discussion has improved from previously, but I still have some doubts on your thesis.

In some parts of the text and tables you refer to ‘right liver lobe’ and ‘left liver lobe’: this is anatomically not correct. Please replace it with either lobes (plural) or division.

Abstract

Line 17: computed tomography, also elsewhere in the text

Introduction

Despite changed from previous version, your introduction is still mainly based on hepatic fibrosis, which is not the focus of your study. You should use 2-3 lines maximum describing the fibrosis and then focus on what you actually did: elasto on normal dogs.

Line 70-71: How can you say these studies failed to reflect the grade of fibrosis? Similarly to your study, these studies are performed on normal dogs and you cannot know the degree of fibrosis, please delete this sentence and consider re-writing the introduction.

Line 80-81: “We hypothesized that 2D-SWE can measure liver stiffness in dogs” how can you test this hypothesis? You base your study on this, without having a gold standard to compare, so you just assume that it works based on previous human and veterinary literature, but this is not a hypothesis of your study.

Methods

Animals: please specify inclusion and exclusion criteria

Line 90: I don’t think a pre-contrast CT scan can exclude liver diseases, I would delete it

Lines 116-117: this is an exclusion criterium, please move to previous section

Lines 141-143: I would also move this information in section 2.2

Line 145: lateral recumbency

CT: lines 164-165: window level and window width? What do you mean with window height? If I interpret it correctly, these are parameters that you change when you visualize the images, they are not acquisition parameters.

Figure 2: the positioning is a normal one, there is no need for this picture. Also Figure 2B does not show much, and it also seemed to be displayed wrong (should be horizontally flipped as the gallbladder is on the left). I would remove this image and if you want to include an image you should include one in which you show the measurements taken.

Results

I still don’t understand the term ‘experimental results’, I think results is enough.

Please always use the same terminology, either anesthesia or anesthetic

Lines 201-202: from the table and figure I understand the values are higher on the right, not on the left. I am also wondering if it is more correct in these cases to refer to p < 0.001 instead of p = 0. The result is the same but I think your p is not exactly 0, is probably so low that the system reads 0 (in some software is still possible to get the actual value).

Lines 213-215: You can avoid repeating the same statement for the two sides, simply say “both on the right and on the left”.

Lines 235-238: the two sentences say exactly the same. You can delete the first one and start with ‘No significant difference’.

Figure 6: as these values are not so different, I think this image does not add much to the table. I would delete it.

Table 4: you can delete this table as all the info are already in the text. In this case is more helpful the figure (7) which gives a visual idea of the results, although not necessary.

Discussion

Lines 278-279: This cannot be the aim of your study, or at least not this one. Please delete this sentence.

Lines 313-322: I still do not understand why the right liver lobes should be more perfused in anesthesia. No other explanation? In my previous comments I asked if the recumbency was the same for each patient: I meant did you always start with the left lobes? Leaving the dog on right recumbency and therefore with the right lobes under the weight of the dog? Could this have influenced your results?

Lines 341-347: I would delete this paragraph as you have a section called conclusions.

Conclusions

This should be a sentence with the main results, it is too long at the current state. Please summarize (you could use lines 341-347 in this section and delete the rest, which is redundant).

Author Response

Dear Reviewer,

Thank you for your valuable feedback on our manuscript. We have carefully reviewed our manuscript and made revisions based on your suggestions. We've answered each of your questions in detail in the attached Word document. We hope you are satisfied with our responses and revisions. We appreciate your cooperation and wish you success in your endeavors.

sincerely

Deng yifeng;dengyif@njau.edu.cn

Department of Veterinary Clinical Sciences, College of Veterinary Medicine, Nanjing Agricultural University, Nanjing, China
